# Single-Cell RNA Sequencing on Formalin-Fixed and Paraffin-Embedded (FFPE) Tissue Identified Multi-Ciliary Cells in Breast Cancer

**DOI:** 10.3390/cells14030197

**Published:** 2025-01-29

**Authors:** Silvia González-Martínez, José Palacios, Irene Carretero-Barrio, Val F. Lanza, Mónica García-Cosío Piqueras, Tamara Caniego-Casas, David Hardisson, Isabel Esteban-Rodríguez, Javier Cortés, Belén Pérez-Mies

**Affiliations:** 1“Contigo Contra el Cáncer de la Mujer” Foundation, 28010 Madrid, Spain; silviagonzalezmartinezbio@gmail.com (S.G.-M.); javier.cortes@maj3.health (J.C.); 2Molecular Pathology of Cancer Group, Ramón y Cajal Health Research Institute (IRYCIS), 28034 Madrid, Spain; irene.carretero.barrio@gmail.com (I.C.-B.); monica.garciacosio@salud.madrid.org (M.G.-C.P.); tamara880723@hotmail.com (T.C.-C.); 3Centre for Biomedical Research in Cancer Networks (CIBERONC), Carlos III Health Institute, 28029 Madrid, Spain; david.hardisson@uam.es; 4Department of Pathology, Ramón y Cajal University Hospital, 28034 Madrid, Spain; 5Faculty of Medicine, University of Alcalá, 28801 Madrid, Spain; 6Centre for Biomedical Research in Infectious Diseases Networks (CIBERINFEC), Carlos III Health Institute, 28029 Madrid, Spain; valfernandez.vf@gmail.com; 7UCA-GTB Unit, Ramón y Cajal Health Research Institute (IRYCIS), 28034 Madrid, Spain; 8Department of Pathology, Hospital Universitario La Paz (IdiPAZ), 28046 Madrid, Spain; isaerodriguez@yahoo.es; 9Faculty of Medicine, Universidad Autónoma de Madrid, 28049 Madrid, Spain; 10International Breast Cancer Center (IBCC), Pangaea Oncology, Quiron-Salud Group, 08017 Barcelona, Spain; 11Medica Scientia Innovation Research, 08007 Barcelona, Spain; 12Medica Scientia Innovation Research, Ridgewood, NJ 07450, USA; 13Department of Medicine, Faculty of Biomedical and Health Sciences, European University of Madrid, 28670 Madrid, Spain; 14IOB Institute of Oncology Madrid, Hospital Beata María Ana de Jesús, 28007 Madrid, Spain

**Keywords:** single-cell RNA sequencing, breast cancer, multi-ciliated cells, fixed fresh tissue, FFPE tissue, tumor heterogeneity

## Abstract

The purpose of this study was to evaluate the suitability of formalin-fixed and paraffin-embedded (FFPE) samples and fixed fresh (FF) samples for single-cell RNA sequencing (scRNAseq). To this end, we compared single-cell profiles from FFPE and matched FF tissue samples of one invasive carcinoma of no special type carcinoma (invasive ductal carcinoma–IDC) and one invasive lobular carcinoma (ILC) to assess consistency in cell type distribution and molecular profiles. The results were validated using immunohistochemistry (IHC), fluorescence in situ hybridization (FISH), and electron microscopy. Additionally, immune cell proportions identified by IHC were quantified using QuPath and compared to the scRNAseq results. FFPE- and FF-derived libraries demonstrated high-quality sequencing metrics, and cellular heterogeneity was similar. No exclusive cell populations were identified by either approach. The four samples analysis identified six types of epithelial cells, as well as tumoral microenvironment populations. The scRNAseq results from epithelial neoplastic cells were concordant with common IHC markers. The proportion of immune cells identified by IHC in FFPE sections were similar to those obtained by scRNAseq. We identified and validated a previously poorly recognized subpopulation of neoplastic multi-ciliated cells (MCCs) (*FOXJ1*, *ROPN1L*). Analysis of FOXJ1 in 214 ER-positive invasive carcinomas demonstrated protein expression in one third of tumors, suggesting frequent focal MCC differentiation. Our results support the suitability of scRNAseq analysis using FFPE tissue, and identified a subpopulation of neoplastic MCC in breast cancer.

## 1. Introduction

The emergence of single-cell RNA sequencing (scRNAseq) technology has revolutionized our understanding of cellular heterogeneity and complexity within tumors, offering unprecedented insights into the molecular mechanisms driving cancer progression and therapeutic resistance [1]. Both tumor and normal scRNAseq studies have been conducted using breast tissue [2,3,4,5,6,7,8,9,10,11]. However, this technology has been exclusively applied to fresh and frozen samples.

There are numerous sample preparation options for single-cell experiments using fresh tissue, frozen tissue, or FFPE tissue. Furthermore, tissue can be dissociated into single cells or single nuclei (snRNAseq). The primary difference between these techniques lies in their approach to sample preparation. scRNAseq is effective for analyzing cells that are easily dissociable and resistant to stress, providing a comprehensive view of cellular function by capturing the complete transcriptome of individual cells. In contrast, snRNAseq is typically preferred for tissues that are difficult to dissociate, and minimizes artificial transcriptional stress responses as compared to scRNAseq [12].

One of the main limitations is that the initial techniques using non-fixed samples relied on the need for rapid processing, making clinical samples very difficult to use. Newer technical approaches allowed for the fixation of fresh and frozen tissues upon collection before proceeding with dissociation, thereby enabling cell storage. However, despite this advantage, the disadvantages and limitations associated with the use of fresh tissue are not entirely resolved. The process still requires a rapid tissue handling pipeline, avoiding prolonged exposure to room temperature conditions, and rapid fragmentation with scalpels to immerse the tissue pieces in the fixation buffer before dissociation.

The utilization of archival formalin-fixed, paraffin-embedded (FFPE) tissues represents a valuable resource for retrospective studies and clinical research. FFPE allows for long-term preservation of tissue specimens, facilitating large-scale retrospective analyses and correlation with clinical outcomes. However, the use of FFPE tissues in scRNAseq analyses poses unique challenges, including RNA degradation and fragmentation, which may influence data quality and interpretation. To date, there are only two comparative studies on the scRNAseq of matched fresh and FFPE samples. One compared samples from three cases of lung cancer [13], and the other compared samples from one case of breast cancer (BC) [14]. These studies provided preliminary evidence of closely correlated transcriptional signatures between samples, although the percentage of detected subpopulations and the individual gene expression varied due to technological differences. Interestingly, FFPE tissue revealed greater cellular diversity compared to fresh tissue samples. Although these studies suggested that single-nucleus transcriptomics of FFPE tissues allows for retrospective analysis of lung tumor cohorts, no studies have yet compared the transcriptomic results derived from whole cells from FF and FFPE tissue in any tumor type.

In this study, we compared scRNAseq profiles of whole cells derived from FF and FFPE tissue from two BC specimens to assess the concordance and differences in cellular composition, gene expression patterns, and molecular signatures between these two sample types. In addition, we evaluated whether scRNAseq data captured conventional immunohistochemical and pathological features, such as the expression of hormone receptors and HER2, and the proportion of immune cells in the tumor. The reliability of the technique was demonstrated by the identification of a subpopulation of neoplastic cells with a gene expression profile typical of multi-ciliated cells (MCCs), which was confirmed by immunohistochemistry (IHC) and electron microscopy. Our findings suggest that retrospective scRNAseq studies using BC archival tissue are reliable, providing useful biological information. In addition, the implication, both biological and clinical, of MCC differentiation in BC deserves further investigation.

## 2. Materials and Methods

### 2.1. Sample Acquisition

Tissue samples were collected from therapy-naïve breast carcinoma tissues from two patients undergoing primary surgery. Informed consent was obtained from all participants before sample acquisition. The ethics committee of Ramón y Cajal University Hospital (Madrid, Spain) approved the use of tissue samples for single-cell gene expression analysis (259-22). Fresh tumor tissues were obtained by a pathologist after breast lumpectomy. Additional tissue, contiguous to the previous sample, was immersed in an OCT compound and snap-frozen in liquid nitrogen for subsequent histological evaluation. The remaining surgical specimen underwent routine histological examination after formalin fixation and paraffin embedding. Paraffin blocks were stored under standard conditions for 1 month. Histological sections were evaluated to select blocks for scRNAseq analysis that included tumor areas that were more similar to those in the frozen sample.

To explore FOXJ1 expression in BC, we selected a series of 214 consecutive ER-positive early-stage invasive breast carcinomas that underwent Mammaprint^®^ analysis for prognostic evaluation.

### 2.2. Pathological and Molecular Characterization of Breast Carcinomas

Histologic typing was performed according to WHO recommendations and cases were graded according to the three-tiered Nottingham histologic grading system. IHC was performed using the BOND-PRIME Polymer DAB Detection System (Leica Biosystems, Wetzlar, Germany) using the antibodies and conditions presented in Appendix B. FOXJ1 expression was analyzed on the ILC and IDC complete slide, and on tissue microarray sections (TMA). TMA was constructed as previously reported [15].

Hematoxylin and eosin (H&E) and IHC slides were digitized in a Philips UFS scanner at 40×. The open-source software QuPath (version 0.5.0) [16] was used for quantification of cells on whole slide images (WSI). The tumor region was manually annotated by a pathologist. Cells within this region were segmented using StarDist (version 0.9.1) [17]. Positive cells for each biomarker were established using a threshold of the mean diaminobenzidine intensity.

Fluorescent In-Situ Hybridization (FISH) on the FFPE section was performed to evaluate the copy number variations of MDM4/1q and HER2/17q loci (Appendix B).

For massive parallel sequencing, 10 sections of 10 μm each were cut per case from the same blocks, from which material was obtained for the scRNAseq technique. Sequencing of DNA was carried out as previously reported [18].

### 2.3. scRNAseq

Fifty milligrams of fresh tissue were fragmented, fixed and dissociated according to the protocol described in Appendix B. For scRNAseq on FFPE tissue, we used 10 tissue sections of 25 μm and followed the protocol described in Appendix B. Single-cell library preparation was conducted following the manufacturer’s protocol for the Chromium Fixed RNA Profiling Reagent Kits for Singleplexed Samples (CG000477 from 10× Genomics). A detailed description of scRNAseq data processing, functional enrichment analysis, and inference of copy number variation (CNV) is shown in Appendix B.

### 2.4. Electron Microscopy

A specimen for electron microscopic examination was obtained from FFPE tissue. The sample was processed, stained and examined according to Mariño et al. [19].

### 2.5. Statistical Analysis

The quantity of main cell types in FF and FFPE samples as well as in IHC images was compared using the paired t test. Differentially expressed genes were identified using the FindAllMarkers function of the Seurat package (version 4) [20,21] with the following parameters: include only positive markers, proportion of expressing cells inside the cluster ≥ 0.1, and difference between proportions of expressing cells inside and outside the cluster ≥ 0.25.

Associations between FOXJ1 expression and clinicopathological variables were analyzed with the Chi test. Statistical analyses were performed using R (version 4.4.2) and SPSS (version 25).

## 3. Results

### 3.1. Clinicopathological and Molecular Features of Tumor Samples

Fresh and matched FFPE tissue samples from two BC patients (Patient 1 and Patient 2) from the Pathology Department of Ramón y Cajal University Hospital (Madrid, Spain) were selected. Patients were diagnosed at age 61 (Patient 1) and 53 (Patient 2) years, respectively. Regarding histological type, the tumor of Patient 1 was an invasive lobular carcinoma (ILC), which had a trabecular pattern growth and was E-cadherin negative. The tumor of Patient 2 was an invasive carcinoma of no special type (invasive ductal carcinoma—IDC) that expressed E-cadherin (Figure 1a and Figure 2). Both tumors were histological grade 2 and were estrogen (ER) and progesterone (PR) receptor positive, but with different expression levels (H-scores: ILC 237.6 ER and 273.6 PR; IDC 56.2 ER and 10.9 PR). Both tumors were HER2-negative and ILC scored +1 and IDC scored 2+ (not amplified by FISH) (Figure 2 and Appendix A). The proliferation index (Ki67) was 15% in the ILC and 18% in the IDC.

Massive parallel sequencing demonstrated *CDH1* (p.Thr515AsnfsTer22) and *PIK3CA* (p.His1047Arg) mutations in the ILC, the two most common mutations in this histological type [22]. The IDC presented an *ERBB2* (p.Leu755Ser) mutation.

### 3.2. Assessment of Single-Cell Transcriptome Quality in Fixed Fresh and FFPE Tissue Samples

The initial analysis of FF and FFPE tissue derived libraries revealed high-quality parameters (Appendix A and Appendix A). Appendix A shows that the majority of cells met the applied quality parameters. Between 0.6% and 3.6% of cells were discarded after applying the quality filters specified in the methodology (Appendix A).

Doublet analysis showed that the proportion of doublets was not related to FFPE processing. In addition, the proportion of reads mapped to the mitochondrial genome, although slightly higher in FFPE samples, was below 20% for the majority of cells across all samples, regardless of origin (Appendix A).

The median number of genes per cell after filtering and doublet removal was not related to the type of sample (FF or FFPE). In fact, it seemed to be more related to the proportion of different cell types in each sample (Appendix A and Figure 1e).

Regarding the median of genes expressed per cell in each cluster independently, discrepancies were observed between FF and FFPE cases in certain populations, such as fibroblasts or epithelial cells (Figure 1e). Nevertheless, these differences cannot be attributed to the fresh or paraffin origin because in some cases, the median was higher in fresh samples (e.g., epithelial 1 and 3), while in other instances it was higher in FFPE samples (e.g., fibroblasts, epithelial cells 2, or epithelial cells 4) (Figure 1e).

Other parameters indicating the high quality of the samples are included in Appendix A and Appendix A.

### 3.3. Cell Heterogeneity and Gene Expression in Fixed Fresh and FFPE Samples

The total number of cells captured from FF tissues was lower than from FFPE samples (21,866 vs. 25,785) (Appendix A). However, the heterogeneity obtained in both types of samples was similar at both a lower (Figure 1c–e) and higher resolution in the sub-analyses of clusters.

Cell populations included five types of epithelial cells: neoplastic epithelial cells 1 to 4, neoplastic MCCs, and normal basal cells. Other cell populations were fibroblasts, endothelial cells, pericytes, lymphocytes, myeloid cells, and mast cells.

There were no populations or subpopulations captured exclusively by one of the approaches (FF or FFPE).

The single-cell data from both FF and FFPE samples were combined into a unified UMAP (Figure 1b), revealing an equal distribution and clusters that shared transcriptome profiles from both tissue types (Figure 1c,d). This suggests that the cell type information remained consistent across the different sample preparation methods (Figure 1b–d and Appendix A).

However, despite observing the same cell types in matched samples, we noted variations in proportions of the type of cells, mainly within IDC samples. In FFPE IDC, fibroblasts predominated; while in FF IDC, epithelial cells 1 and 4 were more prevalent. There were also some variations in ILC, albeit smaller. For example, FF ILC showed a higher percentage of lymphocytes than FFPE ILC, although they were abundant in both sample types (Figure 1e and Appendix A). These results suggested the potential effect of tissue dissociation on cell type quantity.

Regarding gene expression in each cluster, Figure 1f and Appendix A display the expression of canonical markers for each cluster in cells derived from both FF and FFPE tissues separately. The results show consistent percentages of cells expressing the genes and similar average expression levels between matched samples in most clusters. Some differences were observed, particularly in epithelial cell genes, such as *EPCAM*, which presented higher expression in the FFPE samples.

Although minor differences in expression levels may arise due to slight variability in tissue preservation or processing, as well as the fact that the regions analyzed in FF and FFPE samples are adjacent but not identical, cells consistently cluster together, and key overexpressed genes remain the same across both methods.

### 3.4. scRNAseq on FFPE Captures Immunohistochemical and Immune Features of Tumors

After excluding basal cells for further analysis of epithelial cells, we first compared whether the expression of *CDH1*, *ESR1*, *PGR*, *MKI67*, and *ERBB2* obtained through scRNAseq were concordant with the typical immunohistochemical markers used in routine diagnosis (E-cadherin, estrogen and progesterone receptors, Ki67, and HER2). The dot plot of Figure 2b shows increased expression of *CDH1* and *ERBB2* in the IDC and of *ESR1* and *PGR* in the ILC, consistent with IHC results (Figure 2a). Additionally, Appendix A shows that no differences in histological staining are observed when comparing FF and FFPE samples.

We also explored whether immune populations detected by scRNAseq were also detected in similar proportions by IHC. We first automatically annotated the 9964 individual immune cells (lymphocytes, myeloid cells, and mast cells) from the four samples using the Monaco reference dataset from singleR (Figure 3a–c) and obtained the number of cells expressing *CD3*, *CD4*, *CD8*, *MS4A1*, *CD68*, and *KIT* (Figure 3d–f and Appendix A). We then analyzed the protein expression of these genes (CD3, CD4, CD8, CD20, CD68 and KIT) by IHC on FFPE and quantified positive cells digitally on WSIs. Similar results were obtained with both methods (Figure 3f).

ScRNAseq captures the immune microenvironment of both tumors and evidenced a difference in immune infiltrates, which was also observed by IHC (Figure 3). Although ILCs tend to be tumors with a low number of TILs, the tumor we analyzed showed a relatively high number of TILs, mainly due to follicular structures at the periphery of the tumor. It is important to note that this observation appears to be specific to this particular case of ILC and is not representative of ILC tumors in general. Consistently, Narvaez et al. [23] described the organization of TILs in this type of structure in response to immune signals.

### 3.5. scRNAseq Identified Epithelial Cells Heterogeneity Among Neoplastic Cells

A total of 20,039 individual epithelial cells from four samples were analyzed (5058 cells from ILC and 14,981 cells from IDC) (Appendix A). We identified normal basal cells by the expression of specific markers, such as *KRT5*, *TRIM29* and *COL17A1*. This population of cells was present due to normal ducts entrapped in the neoplastic proliferation in both tumors. To confirm that the remaining epithelial cell clusters were neoplastic, we inferred CNVs in these cells using non-malignant cells (immune and basal cells) as a baseline. Figure 4a shows scRNAseq expression of epithelial cells with hallmark chromosome (chr) 1q gain and deletions of 16q and 17p in all populations. Chromosome 17q gain was observed only in the IDC epithelial cell populations. To validate these findings, we analyzed CNVs of *MDM4*/chr 1q and *ERBB2*/chr 17q by FISH (Figure 4b). Therefore, the FISH results validated the utility of scRNAseq data from FFPE samples for inferring tumor CNVs.

We next compared gene expression between ILC and IDC, including all epithelial cell subtypes, and observed differential gene expression between both histological tumor types. As expected, and supporting the good performance of the scRNAseq technique, *CDH1* (E-cadherin gene), some claudins (*CLND3* and *CLND4*), and other genes associated with cell adhesion (*FAT1*) were upregulated in IDC in comparison with ILC. On the other hand, and in agreement with IHC findings, *PGR* was upregulated in ILC (Figure 2b). Interestingly, some genes, such as *GJA1* (the gap junction protein conexin 43) and *IRX2*, which has been reported to be associated with hormone receptor expression in BC, were also upregulated in ILC. Furthermore, we observed a higher expression of *LTF* (lactoferrin), *MUC5B*, and *SCGB2A2* in the epithelial cells of ILC, as reported in normal epithelial breast cells [8] (Figure 5a and Appendix A), which is probably related to a secretory phenotype.

The analyzed ILC was composed of a single cluster of epithelial cells with a homogeneous expression profile (epithelial cells 2). In contrast, the IDC exhibited greater heterogeneity, comprising four distinct subtypes (epithelial cells 1, 3, 4 and MCCs) (Figure 5b–d).

The expression pattern of the most abundant epithelial cells 1 and 3 did not suggest any specific functional differentiation. However, epithelial cells 4 showed higher expression of genes more typical of mesenchymal cells, such as *FBL1*, *FB1*, *CTHRC1* or *COL5A2*, suggesting an epithelial to mesenchymal transcription program in these cells. (Figure 5e and Appendix A).

To further investigate the heterogeneity and differentiation processes within IDC epithelial cells, a cell trajectory analysis was performed. The trajectory plot (Appendix A) highlights the progression and relationships between distinct cellular states. Notably, a distinct branch corresponding to ciliated cells was observed, likely representing a terminal differentiation state. These findings provide insights into the pseudotemporal organization of epithelial cells in IDC.

### 3.6. scRNAseq Identified Neoplastic Epithelial Cells with a Transcriptional Program of Multi-Ciliated Cells

The less abundant epithelial cells in the IDC sample were characterized by the expression of genes related to the ciliary machinery typical of MCC in different normal tissues and tumors, such as fallopian tube [24] and endometrium [25]. Upregulated genes in MCCs included transcription factors involved in MCC fate (*TP63*, *TP73*, *MCIDAS*, *FOXJ1*, *RFX2*), genes involved in centriole amplification (*PLK4*, *CDC20B*, *CCNO*, *DEUP1*), multi-ciliation cell cycle (*E2F7*), centriole dissociation and polarized migration (*CDK1*, *STIL*), and assembly of multiple motile cilia (*CC2D2A*, *RSPH9*, *DZIP1*) (Figure 6c,d and Appendix A).

To confirm the presence of such a population of cells, we performed expression analysis of *TP63* and *FOXJ1* by IHC. *FOXJ1*, the key regulator of the motile ciliogenic program, was only expressed in a subpopulation of cells in IDC (Figure 6a). No FOXJ1 positive cells were observed in the normal epithelial cells or in ILC. The proportion of neoplastic epithelial cells with a MCC transcriptomic program (1.3%), as determined by scRNAseq, was remarkably similar to the proportion of neoplastic cells expressing FOXJ1 by digital analysis on WSI (0.7%).

The presence of MCC was confirmed by electron microscopy (Figure 6b). However, sample fixation affected the image resolution, making it impossible to observe finer cilia details, such as the axoneme structure.

We next evaluated the expression of FOXJ1 in a cohort of 214 ER-positive invasive breast carcinomas, and the clinicopathological features are presented in Appendix A. One third of tumors expressed FOXJ1 in at least 1% of neoplastic epithelial cells. Expression was focal in general and limited to a low percentage of neoplastic cells (mean: 1.36%). We did not observe an association between FOXJ1 positive expression and clinicopathological features (Appendix A). No statistical associations were observed when analyses were performed with a threshold of 5% of FOXJ1 positive cells and separately for ductal and lobular carcinomas.

## 4. Discussion

The results of this study suggested that scRNAseq is a reliable method with both FFPE and FF tissue. Although there were some differences in the results obtained between each sample type, mainly regarding the proportion of cells, both captured the same degree of cellular heterogeneity, as demonstrated by the identification of minor populations of neoplastic cells, such as MCCs.

The differences in cellular populations observed in our study between FF and FFPE samples highlight the impact of sample processing on data outcome. For instance, the higher representation of mesenchymal cells in FFPE samples could be linked to the extended digestion time required by the FFPE protocol, approximately 20 min longer than the fresh tissue protocol. This extended processing time might favor the extraction of certain cell types. Similarly, the FF ILC sample showed a higher proportion of lymphocytes compared to FFPE. This may be due to the rapid processing of fresh tissues, preserving more lymphocytes that typically express fewer genes than other cell types. In agreement with our results, Trinks et al. [13] found that immune cells transcriptomes were enriched, but epithelial and stromal cells transcriptomes were depleted from fresh tissue single-cell libraries in comparison with those obtained from FFPE tissue. In lung tissue, it has been reported that the cell type proportions varied widely between scRNAseq and snRNAseq with a predominance of immune cells in the former and epithelial cells in the later [27].

The observation that in both types of samples we identified the same types of cells, and the concordance with the IHC studies on FFPE sections, support the reliability of both approaches. Thus, regarding the expression of *ESR1*, *PGR* and *ERBB2*, scRNAseq results were concordant with those observed in FFPE sections, confirming the higher expression of ER and PR in ILC and higher expression of *ERBB2* in IDC. Interestingly, this tumor showed an *ERBB2* mutation (p.Leu755Ser) and a gain of one copy at 17q, including the *ERBB2* gene, as demonstrated by sequencing and FISH, respectively (Figure 4). Our findings demonstrated that scRNAseq results from FFPE samples are also highly reliable in detecting CNVs using the R package inferCNV [28]. Although this study, based on only two different tumors, did not intend to evaluate differences between the two main histological types of BCs, we demonstrated the absence of *CDH1* expression by scRNAseq in ILC, concordant with the absence of protein expression demonstrated by IHC.

scRNAseq also captures the immune microenvironment of both tumors and evidenced a difference in immune infiltrates, which was also observed by IHC. Although ILCs tend to be tumors with low immune infiltration, the tumor we analyzed showed a relatively high number of immune cells, mainly due to tertiary lymphoid structures at the periphery of the tumor, which were included in our scRNAseq analysis, and which have been described in up to 60% of breast carcinomas [23].

An important finding in our study was the identification of a subpopulation of epithelial cells in IDC with a transcriptomic program typical of MCCs. The presence of these cells was further validated by electron microscopy (Figure 6b). MCCs are terminally differentiated cells that contain dozens to hundreds of motile cilia and line the airway tracts, brain ventricles, and reproductive ducts. We found that MCCs express genes involved in all stages of multi-ciliary differentiation, from precursor to differentiated cells, as occur in different normal tissues [26] (Figure 6b,c and Appendix A). Thus, we observed overexpression of several transcriptional regulators of multi-ciliogenesis, such as *TP63, MCIDAS*, *TP73*, *FOXJ1* and *RFX2*. Whereas both *MCIDAS* and *TP73*, which is considered as a competence factor for MCC differentiation, regulate the expression of *RFX2* and *FOXJ1*, *MCIDAS* expression also participates in centriole amplification, a process in which *CDC20B* and *CNNO* play an important role [26].

Once the MCC cell fate is determined, these cells have to exit the cell cycle and create a permissive environment for massive centriole production. A recent study has proposed that MCCs use an alternative cell cycle that orchestrates differentiation instead of controlling proliferation. The so-called multi-ciliation cycle omits cell division and chromosome duplication and is regulated by *E2F7*, which was also overexpressed in MCCs in the present study. *E2F7* prevents expression of DNA replication genes in the S-like phase and blocks aberrant DNA synthesis in differentiating MCCs [29].

To form all the motile cilia, hundreds of proteins need to be synthesized in a short period and cooperate to establish a precise and complicated arrangement. Strong experimental evidence has established FOXJ1 as the master regulator of the motile ciliogenic program. The key role of FOXJ1 in the specification of the motile cilia has been so well established that the term FIG has been coined to specify the FOXJ1-induced genes [29,30]. Importantly, the role of FOXJ1 in directing ciliogenesis is strictly restricted to motile cilia, in contrast to other TFs, such as RFX2, which are also involved in the regulation of primary cilia. It is important to mention that the MCC identity is inherently labile, as its maintenance requires constant *FOXJ1* transcriptional activity [31].

To the best of our knowledge, only two previous ultrastructural studies in the 1980s described the presence of MCC in occasional BCs [32,33]. No further studies have reported this type of cells in normal breast or BC, nor has its biological significance been evaluated. However, three previous studies analyzing the same TCGA dataset, searching for potential prognostic factors, have reported the expression of *FOXJ1* mRNA as a favorable prognostic factor in breast cancer [34,35,36]. The authors of these three similar studies did not consider the expression of *FOXJ1* in the context of MCC differentiation and did not propose any interpretation of this finding. Taking into account that FOXJ1 is highly specific of MCCs, and that we observed a good concordance between the number of MCCs detected by scRNAseq and the number of cells expressing FOXJ1 by IHC, we performed a preliminary study of FOXJ1 expression by IHC in order to evaluate the frequency and possible significance of MCC differentiation in BC. To this end, we selected a cohort of luminal breast carcinomas in which Mammaprint^®^ results for prognostic evaluation were available. In this selected group of cases, we detected FOXJ1 expression in at least 1% of cells in one third of tumors, but expression was generally limited to a low percentage of cells (median: 0) (Appendix A). In this series of luminal tumors, FOXJ1 expression was not associated with clinicopathological factors, such as age, stage, histological type, tumor grade, or risk, as evaluated by Mammaprint^®^. In contrast to normal breast, MCC differentiation occurs in normal fallopian tube [24] and endometrium [25]. Moreover, FOXJ1 expression has been reported to be associated with a favorable prognosis in high grade serous carcinomas [37] and endometrial carcinomas [38].

The limitations of this study regarding scRNAseq include the analysis of only two tumors and the absence of additional types of samples, such as fresh tissue or single nuclei. In addition, we only tested paraffin blocks with a limited period of storage (one month). Regarding the analysis of MCC differentiation in BC, the main limitations were the study of only FOXJ1 as a marker of multiciliation, the use of TMA sections, and the analysis of only luminal carcinomas.

## 5. Conclusions

This proof-of-concept study found that scRNAseq analysis of FFPE breast carcinomas, subjected to a limited period of storage, is feasible and recapitulates common pathological and immune features of tumors. In addition, we identified the presence of MCCs in BCs. Further studies comparing a larger number of samples and analyzing different periods of archive time are required. Moreover, future studies should analyze FOXJ1 expression and other markers of MCC differentiation in a large series of breast carcinomas, including all molecular subtypes, to better understand the biological and clinical significance of this specific type of cellular differentiation.

## Figures and Tables

**Figure 1 cells-14-00197-f001:**
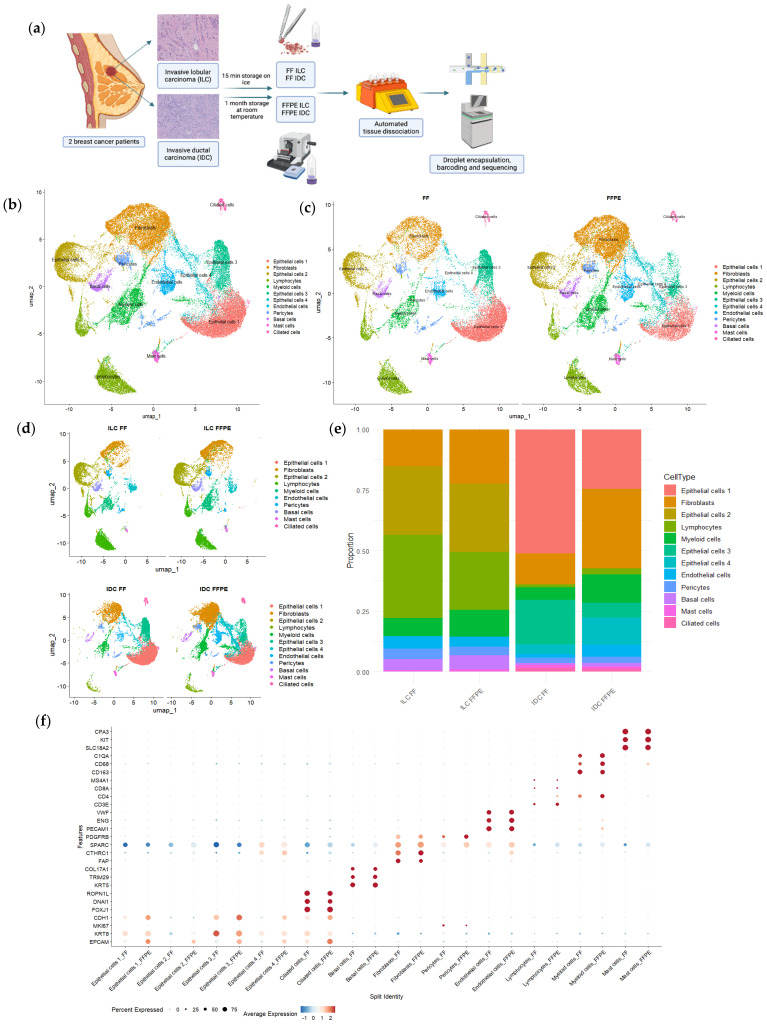
Process of single-cell analysis in fresh fixed and FFPE tissue samples and cell heterogeneity and gene expression in both type of samples. (**a**) Workflow of tissue specimens used for the study (created in BioRender.com). (**b**) UMAP plot manually annotated by cell type using canonical type-specific marker genes. (**c**) UMAP plot split by fresh and FFPE sample origins (**d**) UMAP plot split by sample ID. (**e**) Bar plot showing cellular proportions in each sample. (**f**) Dot plot representing the percentage of cells and the average gene expression of canonical markers for each cluster in fresh and FFPE samples.

**Figure 2 cells-14-00197-f002:**
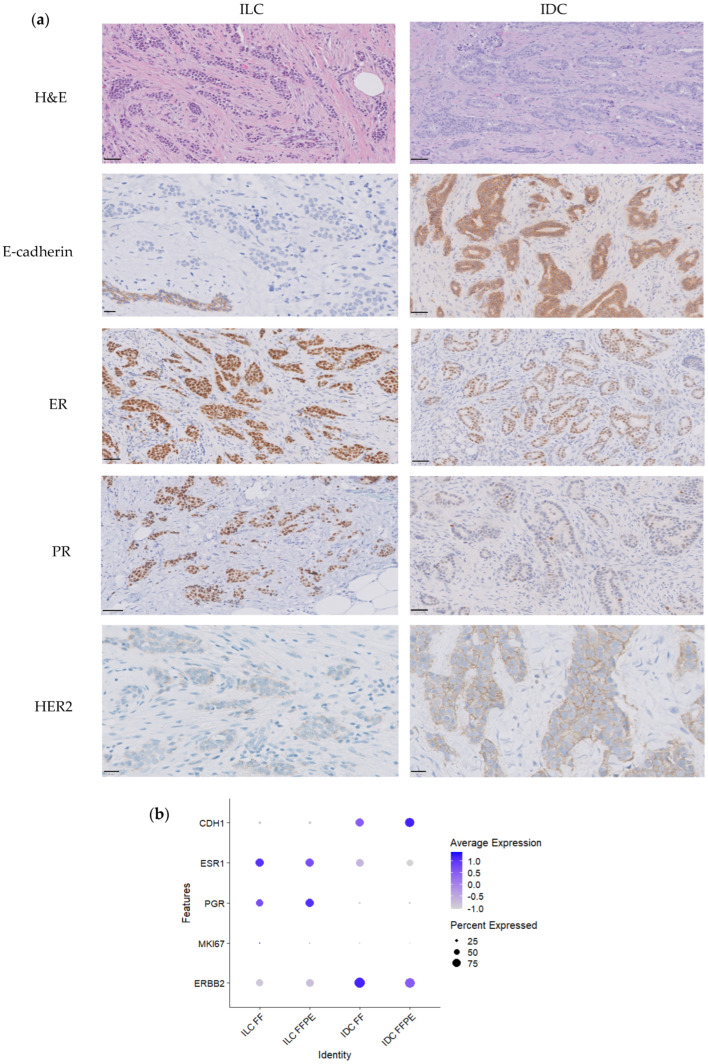
Hematoxylin and eosin staining, and marker expression evaluation used in routine diagnosis. (**a**) H&E, E-cadherin, ER, PR, and HER2 in ILC and IDC tumors (scale bar 50 μm). (**b**) Dot plot showing the expression of *CDH1*, *ESR1*, *PGR*, *MKI67*, and *ERBB2* obtained through scRNAseq.

**Figure 3 cells-14-00197-f003:**
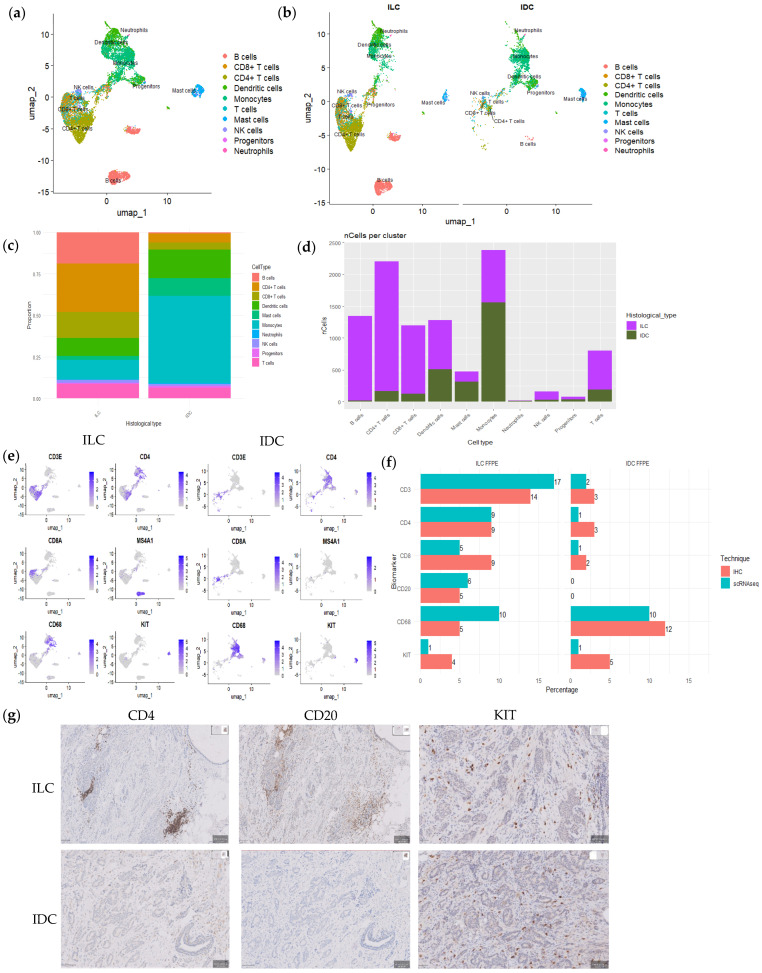
Immune cell scRNAseq and immunohistochemistry analysis. (**a**) UMAP plot automatically annotated by immune cell type using the Monaco reference dataset from singleR. (**b**) UMAP plot split by histological type. (**c**) Bar plot showing immune cellular proportions in each histological type. (**d**) Bar plot showing the number of each type of immune cells, colored by histological type. (**e**) Feature plot displaying specific markers of immune cells per histological type. (**f**) Graph showing the percentages of each immune population per sample by scRNAseq and digital analysis relative to the total number of cells in the sample. (**g**) Immunohistochemistry of FFPE ILC and FFPE IDC showing CD4, CD20 and KIT expression in immune cells (scale bar 50 μm).

**Figure 4 cells-14-00197-f004:**
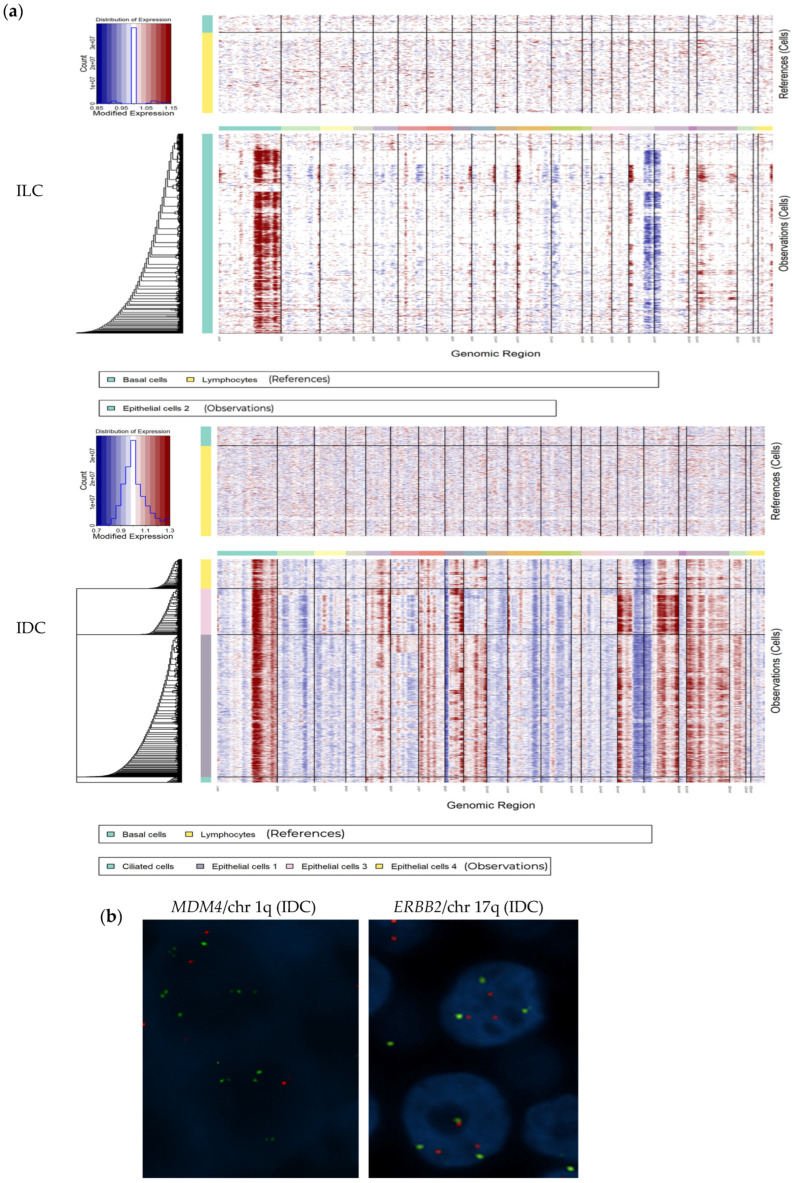
Epithelial cell CNV analysis. (**a**) Expression-based inference of the CNV landscape across patient-wide malignant epithelial cells. The heatmap obtained from the inferCNV tool depicts the putative CNV landscape across five epithelial cells while considering healthy basal cells and lymphocytes as a reference. (**b**) FISH of FFPE IDC showing chr 1q amplification and chr 17q gain (green signals represent gene-specific probes, while red signals correspond to centromeric probes).

**Figure 5 cells-14-00197-f005:**
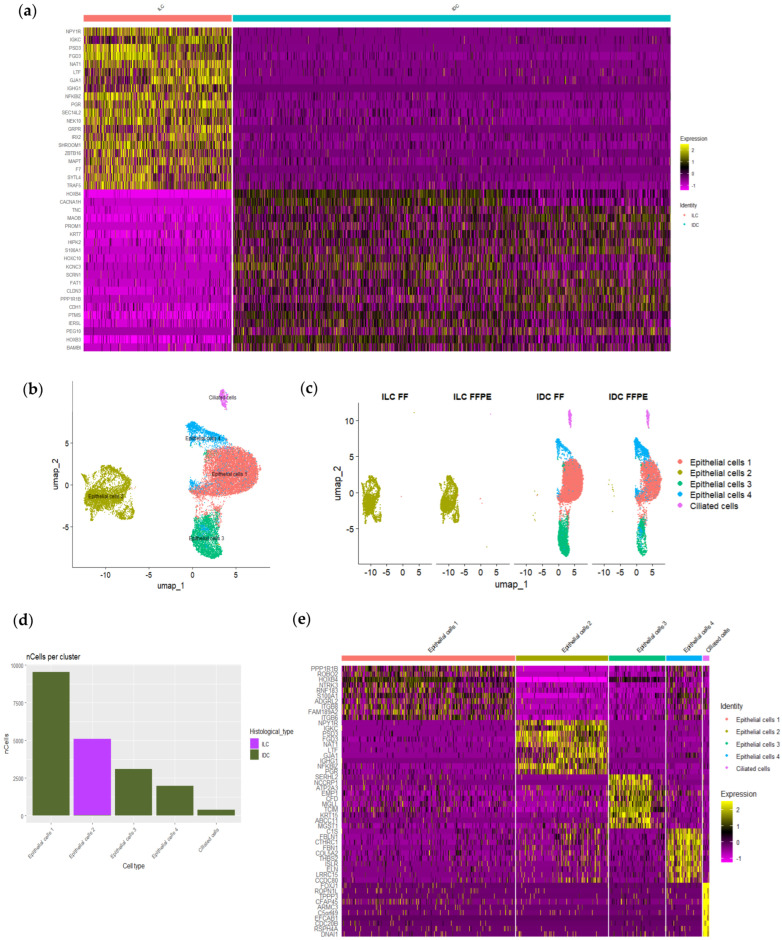
Epithelial cell scRNAseq analysis. (**a**) Heatmap representing the top 20 most overexpressed genes (avg_log2FC > 1) in ILC vs. IDC epithelial cells. (**b**) Annotated UMAP plot of 20,046 individual epithelial cells from four samples generated from the top 30 principal components of all single-cell transcriptomes integrated. (**c**) UMAP plot split by sample ID. (**d**) Bar plot showing the number of each type of epithelial cells, colored by histological type. (**e**) Heatmap representing the top 10 most overexpressed genes (avg_log2FC > 1) in each epithelial cell population.

**Figure 6 cells-14-00197-f006:**
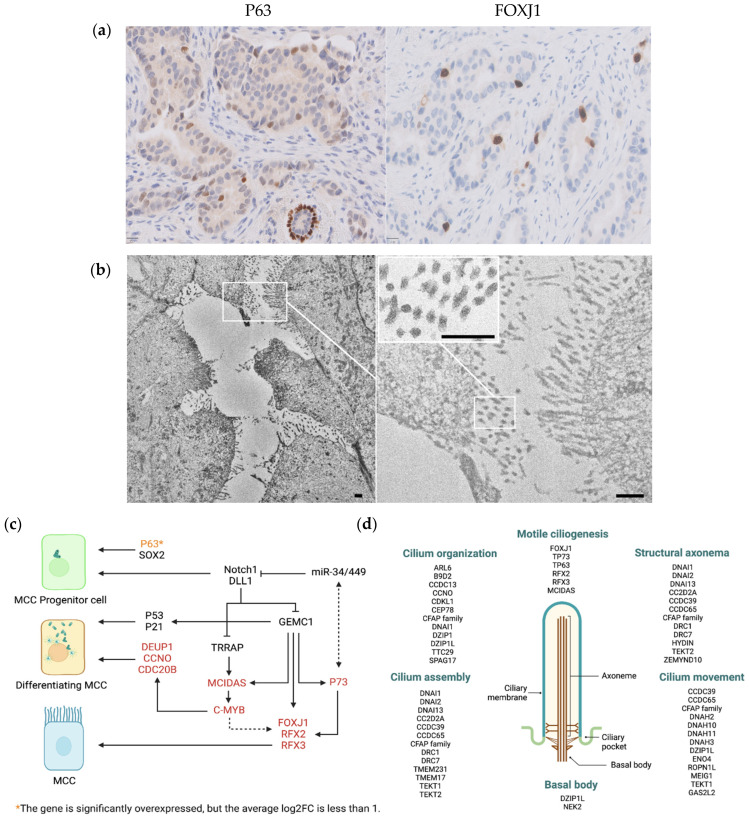
Multi-ciliated cells in the IDC case. (**a**) Immunohistochemistry of FFPE IDC showing P63 and FOXJ1 expression (scale bar 20 μm). (**b**) Scanning electron microscopy image showing the dense and organized structure of cilia on the surface of MCC (scale bar 500 nm). (**c**) Regulation of MCC cell fate determination, highlighting the genes overexpressed in the MCC population from our study (figure adapted from Lyu et al. [26]) (created in BioRender.com). (**d**) Genes involved in the regulation, formation, and function of cilia according GO-enriched biological processes indicated in Appendix A (created in BioRender.com).

## Data Availability

ScRNAseq data from this study are available through the Gene Expression Omnibus under accession number GSE278793. Code related to the analyses in this study can be found on GitHub at: https://github.com/Gonzalez-Martinez/Single-cell-RNA-seq-on-FF-and-FFPE-tissues.git (accessed on 25 January 2025).

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
