# Peer review of "Single-Cell RNA Sequencing on Formalin-Fixed and Paraffin-Embedded (FFPE) Tissue Identified Multi-Ciliary Cells in Breast Cancer"

_cells, 2025, doi:10.3390/cells14030197_

Round 1
Reviewer 1 Report
Comments and Suggestions for Authors
This study presents the results of a comparative analysis of expression profiles between formalin-fixed, paraffin-embedded (FFPE) and fresh frozen (FF) samples for single-cell RNA sequencing. Specifically, the analysis compares invasive ductal carcinoma (IDC) and invasive lobular carcinoma (ILC).
Overall, the experiment and analysis were well organized, and the manuscript is well described. However, several major issues need to be addressed for clarity and comprehensiveness.
In Figure 1f, markers such as EPCAM exhibit differences in expression between FFPE and FF samples within specific cell types. It would be beneficial to include a detailed explanation of the underlying reasons for these differences.
Additional explanation is needed regarding the interpretation of the results in Figure 3d. Why are there a significantly higher number of immune cells, including most immune cell types, in ILC compared to IDC? Possible biological or technical reasons should be explored.
Figure 6d lacks any accompanying explanation in the text. Furthermore, a reference for the source of the listed markers is necessary.
The manuscript states that FOXJ1 is expressed in a small fraction of the cell population and in multiciliated cells (MCC). However, it is unclear how this detection aligns with the analysis of The Cancer Genome Atlas (TCGA) breast cancer dataset, which relies on bulk RNA sequencing data. Bulk sequencing lacks the resolution to identify small cell populations effectively.
Moreover, it should be noted that TCGA data represents fresh frozen samples, which are different from FFPE samples analyzed in this study. This discrepancy raises concerns about the direct comparability of FOXJ1 expression profiles between the datasets.
Reviewer 2 Report
Comments and Suggestions for Authors
In the manuscript titled "Single-cell RNA sequencing on formalin-fixed and paraffin-embedded (FFPE) tissue identified multi-ciliary cells in breast cancer," Gonzalez-Martinez et al. explore two main subtypes of invasive breast cancer: invasive ductal carcinoma (IDC) and invasive lobular carcinoma (ILC). Single-cell transcriptome analysis has significantly advanced our understanding of various cancers, including breast cancer. In this study, the authors performed single-cell RNA-seq analysis on the two subtypes from two patients and compared two commonly used tissue processing methods: FFPE and fresh-frozen. The primary aim of the authors was to identify the contribution of these methods to batch effects in single-cell RNA-seq analysis of patient samples. This is an important consideration in the design of such studies, as tissue processing greatly influences the conclusions drawn. For example, in complex tissues like the brain, differences in neuron proportions have been observed depending on tissue fixation methods and whether single-cell or single-nuclei preparations are used. The authors began by characterizing the tissues using clinical pathology, identifying defining histological signatures and molecular characteristics, including known mutations. After conducting necessary quality checks, they annotated and identified various cell types in the two different subtypes and two different sample processing methods. Interestingly, they found differences between epithelial cell types and their proportions across the two subtypes, but not between the two processing conditions. They also identified differences in immune cell type proportions when comparing the two subtypes. Finally, they discovered a multi-ciliated cell (MCC) signature in a subpopulation of cells in the IDC subtype, indicating activation of distinct differentiation programs in the two subtypes of breast cancer.
Overall, the manuscript is well written, and the methods used are well presented with necessary details.
I have a few minor comments and suggestions,
1) The embedded figure labels especially in the UMAPs are hardly visible, making it difficult to identify various cell types.
2) In Fig.1 the authors show the various cell type markers/genes comparing the Fresh frozen (FF) and Formalin-fixed (FFPE)in both the breast cancer subtypes. It would be interesting to show the differences or similarities between the two processing methods in each subtype separately. Was there a difference observed.
3) Please present the scales for the images in Fig.2 a
4) Do the authors observe differences in histological staining when comparing FF vs FFPE?
5) Is there a reason for using 20% cut-off for fraction of mitochondrial reads?
6) Why did the authors use log normalization over SCTransform , which seems to be a more robust method of normalization for single-cell analysis?
7) Authors have used basal cells, annotated as “normal” based on few cell-type markers and used them as reference for inferCNV analysis. Is there a reason for this? Why have they not used endothelial cells and immune cells or external references instead? Is there a possibility for existence of basal-like tumor cells in IDC and ILC subtype?
Reviewer 3 Report
Comments and Suggestions for Authors
Dear authors,
Thank you for submitting your work. After reviewing your manuscript, I understand that the proposed study focuses on evaluating the performance of scRNA-seq using FFPE samples compared to FF samples. This work is well-grounded, as all computational biology findings were supported by additional experimental validations. For instance, scRNA-seq findings were corroborated by IHC and electron microscopy results, while CNV analyses were validated using FISH. Moreover, this study identified and investigated the presence of MCCs in BCs. While I have no major concerns regarding the manuscript, there are several areas where improvements can be made:
- Grammar and typos: Please proofread the manuscript carefully to address the following:
- Line 33: Replace "especial" with "special."
- Line 43: Replace "IHQ" with "IHC."
- Line 58: Change the comma to a period.
- Line 224: "Plot" should start with a lowercase letter.
- Line 259: Remove the extra parentheses.
- Line 358: Replace "scRNaseq" with "scRNAseq."
- Line 377: Italicize "CNNO."
- Line 416: Add the missing space.
- Lines 283–284: Address the minor grammar issue.
- GitHub link: The provided GitHub link is empty.
- The legend for Figure 4 needs improvement. It is confusing that both baseline cells and epithelial cells share the same color. Additionally, please clarify the meaning of the rainbow range of the horizontal cluster coloring.
- Figure quality: Please at least improve the resolution of Figures 1, 3, 4, S1c, and S1d. Note that the title of Figure 6a is currently overlaid by the plot and needs adjustment.
- Epithelial cell clusters: Regarding the four epithelial cell clusters identified in IDC samples, could you further annotate their subtypes, potentially incorporating trajectory analysis?
Thank you for your attention to these matters.
Reviewer 4 Report
Comments and Suggestions for Authors
The authors have performed a single-cell transcriptomics study on the FF and FFPE tissue samples of an ILC patient and an IDC patient. The main research goals of comparing scRNAseq profiles between these two sample types and elucidating the carcinoma's molecular characteristics are outlined, and they have considerable promise for enhancing our comprehension of breast cancer. Although, this manuscript is very well-structured with respect to utility, there are critical issues that undermine the validity and generalizability of the findings. Therefore, I regret to inform you that my recommendation is for rejection.
FFPE tissues are typically subjected to snRNAseq due to the challenges associated with dissociating tissues and the ability to obtain higher quality sequencing data. It is crucial for the authors to delineate a comprehensive comparison between scRNA-seq and snRNAseq methodologies. Moreover, the authors should encompass not only the data derived from the present study but also integrate findings from previous studies on ILC and IDC. Such a comparative analysis will provide valuable insights into the strengths and limitations of each approach when applied to FFPE tissues, thereby strengthening the methodological foundation of the research.
The study's conclusion is based on data from only two individuals, each representing ILC and IDC. This sample size is grossly inadequate to draw any meaningful conclusions about the molecular features of these two distinct types of breast cancer. The limited data cannot capture the inherent heterogeneity and variability within each cancer type, nor can it account for inter-individual differences that may significantly impact the scRNAseq profiles. The authors must acknowledge this glaring limitation and refrain from making definitive statements based on such a small sample. A substantially larger and more diverse cohort is necessary to validate the findings and ensure that the conclusions are robust and applicable to the broader patient population.
In section 3.6, the authors have identified the expression of TP63 and FOXJ1 in IDC MCCs. However, the function of FOXJ1 appears to be limited in these carcinomas. The authors need to clarify the significance of this identification. If FOXJ1's role is indeed restricted, further analysis of other genes and the importance of MCC should be conducted. Moreover, the scRNAseq data reveal a cluster of ciliated cells in ILC, which has not been analyzed by the authors. A comprehensive examination of these cells' features is warranted to provide a more holistic understanding of the cellular landscape in ILC.
Page6&14,Figure 1b-e和Figure 5b-c:Discrepancies in the clustering of epithelial cells from ILC and IDC are observed in Figures. The authors should elucidate the specific statistical analysis methods utilized for clustering and provide a thorough explanation of the current results. It is essential to determine whether the clustering strategies need to be harmonized to achieve consistency across the figures, thereby ensuring the accuracy and comparability of the research outcomes.
Page5-6, Figure 1f & line 215-218: The description lacks clarity regarding which sample's results are being presented—ILC, IDC, or a combination of both. The authors are required to provide further explanation and clarification. If necessary, the figure should be redrawn to ensure that the information is accurately conveyed.
Comments on the Quality of English Languageneed to be approved.
Round 2
Reviewer 1 Report
Comments and Suggestions for Authors
The authors have addressed the comments and concerns and I have no futher comments.